# A Combination of Microarray-Based Profiling and Biocomputational Analysis Identified miR331-3p and hsa-let-7d-5p as Potential Biomarkers of Ulcerative Colitis Progression to Colorectal Cancer

**DOI:** 10.3390/ijms25115699

**Published:** 2024-05-23

**Authors:** Pilar Chacon-Millan, Stefania Lama, Nunzio Del Gaudio, Antonietta Gerarda Gravina, Alessandro Federico, Raffaele Pellegrino, Amalia Luce, Lucia Altucci, Angelo Facchiano, Michele Caraglia, Paola Stiuso

**Affiliations:** 1Department of Precision Medicine, University of Campania “Luigi Vanvitelli”, 80138 Naples, Italy; pilar.chaconmillan@unicampania.it (P.C.-M.); stefania.lama@unicampania.it (S.L.); nunzio.delgaudio@unicampania.it (N.D.G.); antoniettagerarda.gravina@unicampania.it (A.G.G.); alessandro.federico@unicampania.it (A.F.); raffaele.pellegrino@unicampania.it (R.P.); amalia.luce@unicampania.it (A.L.); lucia.altucci@unicampania.it (L.A.); michele.caraglia@unicampania.it (M.C.); 2Biogem Scarl, Institute of Genetic Research, Laboratory of Molecular and Precision Oncology, 83031 Ariano Irpino, Italy; 3Institute of Experimental Endocrinology and Oncology “Gaetano Salvatore” (IEOS)-National Research Council (CNR), Via Sergio Pansini, 80131 Naples, Italy; 4Programma di Epigenetica Medica, A.O.U. “Luigi Vanvitelli”, 80138 Naples, Italy; 5Institute of Food Sciences, National Research Council, 83100 Avellino, Italy; angelo.facchiano@cnr.isa.it

**Keywords:** competing endogenous RNA, ulcerative colitis, colorectal cancer, miRNA, biomarkers inflammatory bowel disease

## Abstract

Ulcerative colitis (UC), an inflammatory bowel disease (IBD), may increase the risk of colorectal cancer (CRC) by activating chronic proinflammatory pathways. The goal of this study was to find serum prediction biomarkers in UC to CRC development by combining low-density miRNA microarray and biocomputational approaches. The UC and CRC miRNA expression profiles were compared by low-density miRNA microarray, finding five upregulated miRNAs specific to UC progression to CRC (hsa-let-7d-5p, hsa-miR-16-5p, hsa-miR-145-5p, hsa-miR-223-5p, and hsa-miR-331-3p). The circRNA/miRNA/mRNA competitive endogenous RNA (ceRNA) network analysis showed that the candidate miRNAs were connected to well-known colitis-associated CRC ACVR2A, SOCS1, IGF2BP1, FAM126A, and CCDC85C mRNAs, and circ-SHPRH circRNA. SST and SCARA5 genes regulated by hsa-let-7d-5p, hsa-miR-145-5p, and hsa-miR-331-3p were linked to a poor survival prognosis in a CRC patient dataset from The Cancer Genome Atlas (TCGA). Lastly, our mRNA and miRNA candidates were validated by comparing their expression to differentially expressed mRNAs and miRNAs from colitis-associated CRC tissue databases. A high level of hsa-miR-331-3p and a parallel reduction in SOCS1 mRNA were found in tissue and serum. We propose hsa-miR-331-3p and possibly hsa-let-7d-5p as novel serum biomarkers for predicting UC progression to CRC. More clinical sample analysis is required for further validation.

## 1. Introduction

UC is an IBD subtending self-sustained relapsing-remitting colorectal inflammation limited to the mucosal surface with a significant and rising epidemiological burden [1]. UC pathogenesis is particularly complex and not yet fully elucidated, resulting from the interaction between genetic factors, genomic and epigenomic factors, the mucosal immune system, and gut microbiota [2]. The imbalance in immune homeostasis towards a pro-inflammatory status results in UC patients having a significantly higher risk of CRC at ten and twenty years from UC diagnosis compared to the general population [3]. In detail, several IBD-related factors behave as independent risk factors of UC-CRC (i.e., pancolitis, young age at UC onset, long-standing disease, associated primary sclerosing cholangitis as a family history of CRC) [4]. In other words, a more extensive disease and a longer UC duration are the two pillars on which the risk of CRC in a UC patient is built. On these bases, it is imperative to set up adequate endoscopic surveillance of CRC in UC patients for an early detection of colonic dysplasia [5].

Long-lasting inflammation of the intestinal mucosa directly impacts the risk of CRC through the chronic activation of several pro-inflammatory pathways and by increasing oxidative stress [6,7]. The latter results in reactive oxygen species genesis, causing DNA damage and potentially affecting critical genome sequences for cell cycle regulation (such as p53 or APC genes) [6]. Cytokine pathways contribute to this pro-neoplastic transition via their IBD-related activation. The most common altered pathways leading to cancer development in UC are NF-κB, IL-6/STAT3, IL-23/T_H_17, COX-2/PGE_2_, and Wnt/β-Catenin [6,8].

In this complex network of the molecular interactions increasing UC-CRC risk, the role of epigenetics is gradually emerging. The regulation of inflammatory processes is also mediated by noncoding miRNAs post-transcriptionally regulating protein expression through translation block of messenger RNAs (mRNAs) [9]. In this respect, several miRNAs have been associated with the pathogenesis of UC-CRC (i.e., miR-18a, 19a, 21, 26b, 31, 146b, 155, and others) [10]. Also, gut microbiota can, in turn, interfere with miRNA regulation as in the case of enterotoxigenic *Bacteroides fragilis*, which is able to promote pro-neoplastic transition by the downregulation of miR-149-3p [11].

The aim of this pilot study was to identify serum predictive biomarkers of the progression from ulcerative colitis to colorectal cancer using a combined approach based on a miRNA microarray profiling and a biocomputational analysis.

## 2. Results

### 2.1. Patient Characteristics: Nonrelated IBD-Inflammation, UC, and CRC Patients

We enrolled a total of 10 patients per group (nonrelated IBD inflammation, UC, and CRC). Table 1 summarizes the clinical parameters of each group. Briefly, UC patients coursed the disease for at least 10 years, a condition that increases CRC development risk. We applied the TNM classification for staging CRC disease. Notably, all the groups were balanced for sex, age, and stage of disease.

### 2.2. In Serum Circulating miRNA Expression Profiling of Nonrelated IBD-Inflammation, Ulcerative Colitis, Colorectal Cancer Patients

Low-density miRNA microarray analysis was carried out to compare the serum miRNA expression profiles of the CTR, UC, and CRC groups. We observed that endogenous hsa-miR-483-5p is stably expressed in all three groups. Therefore, it was used to normalize the microarray profile. Thereafter, to provide a quick view of the UC and CTR miRNA expression profiles compared to the CRC serum samples, a hierarchical clustered heatmap was created (Figure 1A). We identified the top five downregulated miRNAs (hsa-miR-122-5p, hsa-miR-139-5p, hsa-miR-146b-5p, hsa-miR-148a-3p, and hsa-miR-150-5p) and the top five upregulated miRNAs (hsa-let-7d-5p, hsa-miR-16-5p, hsa-miR-145-5p, hsa-miR-223-3p, and hsa-miR-331-3p) in CRC patients vs. UC patients (Ct < 32, FC ≥ |2|).

Additionally, we compared the expression of our miRNA candidates identified in UC-CRC progression with their expression in nonspecific IBD-related inflammation progression to CRC. Only the upregulated candidates showed a unique UC inflammatory profile and were forward analyzed (Figure 1B).

### 2.3. CeRNA Network Revealed the Crosslink between circRNAs, miRNAs, and mRNAs

To analyze the interconnections between miRNAs, a circRNA-miRNA-mRNA ceRNA network was built. Our top 5 upregulated miRNA candidates paired into a total of 2 circRNAs, forming 2 miRNA-circRNA connections and 549 mRNAs, generating 567 miRNA-mRNA links (Figure 2A). The top 5 upregulated miRNAs were interconnected through 64 links to 32 mRNAs and to 2 circRNAs by 2 connections. The results revealed that hsa-miR-16-5p, hsa-miR-145-5p, hsa-miR-223-3p, and hsa-let-7d-5p shared a common predicted target: ACVR2A mRNA. Interestingly, hsa-miR-16-5p, hsa-miR-145-5p, and hsa-let-7d-5p shared several predicted targets with hsa-miR-331-3p (CCDC85C, TSPAN18, SOCS1, MYRIP, IGF2BP1, and FAM126A mRNAs). Finally, hsa-miR-145-5p and hsa-miR-331-3p were targeted by circ-CEP128 and circ-SHPRH [12], respectively. The detailed results from the target analysis are reported in Appendix A. Considering the highest network connections, a further analysis was performed using the top five upregulated miRNA candidates. A ceRNA subnetwork showed the most relevant interactors with the top upregulated miRNAs (Figure 2B).

### 2.4. PPI Network Analysis Showed Protein Clusters and Top Hub Genes Involved in the Tumoral Progression Landscape

To create a comprehensive picture of the tumoral landscape, we constructed a PPI network to identify interactions between the mRNAs targeted by the upregulated miRNA candidates. The analysis highlighted that the previously shown 549 mRNAs (Figure 2A) were in silico codified into a total of 526 proteins [12]. The ID protein members of this network are described in Appendix A. The PPI network was subjected to an additional investigation, to identify 14 protein clusters (Appendix A) with a network scoring degree cut-off > 2. The relationships between the different clusters were visually distinguished by the density of their connecting edges (Figure 3A). The results showed that cluster 1 and cluster 2 were the most interconnected groups, with 28 nodes and a total of 31 links. Additionally, the top ten hub gene analysis was carried out to identify the molecules with the highest interaction rate in the PPI network. Our results show the top ten hub genes: TP53, VEGFA, CASP3, SMAD4, IGF, SMAD2, TGFBR1, THBS1, COL1A1, and FOXO1 genes (Figure 3B). Protein clusters that presented the top ten hub genes were also identified: TP53, CASP3, VEGFA, and FOXO1 formed part of cluster 1 (Figure 3C), whereas SMAD2, SMAD4, IGF1, TGFBR1, COL1A1 and THBS1 were members of cluster 2 (Figure 3D). Notably, cluster 5 (Figure 3E) contained the previously mentioned ACVR2A protein that was targeted by let-7d-5p, hsa-miR-16-5p, hsa-145-5p, and hsa-miR-223-3p (Figure 2B).

### 2.5. Functional Enrichment and Pathway Analysis of the PPI Network Members Pointed towards the Classical Colorectal Tumoral Axis and a Neuronal Component

To examine the functional role of our miRNA-related PPI in CRC progression from UC disease, we analyzed the 526 PPI network members by GO. Among the biological processes, we found enriched terms related to a neuronal component, such as neurotransmitter levels, transport and secretion, and the regulation of synapsis (Figure 4A). To the same extent, the cell component term analysis also included the neuronal cell body and distal axon, presynaptic membrane, and tumoral-related terms such as cell–cell junction (Figure 4B). Finally, SMAD and phosphatase binding were identified as molecular function terms (Figure 4C). To understand the role of our miRNA-related protein members, we then performed KEGG pathway analysis. The top 10 KEGG pathways involving our miRNA-related protein patterns showed as enriched axes the proteoglycans in cancer, cellular senescence, colorectal cancer, and neurotrophin signaling, among others (Figure 4D). Subsequently, to more deeply characterize the miRNA-affected genes in the classical CRC axis, we deeply analyzed the above-reported CRC KEGG term by generating a DAVID pathway map (see Appendix A). This map revealed that the predicted genes affected PI3K-Akt signaling, apoptosis, the apoptosis-dependent WNT pathway by the altered expression of the Bcl-2, CASP3, PI3K, PKB/Akt genes, and the MAPK signaling pathway through dysregulating the expression of the Ral gene. The TGF-β pathway was also affected by the abnormal expression of TGFβRI, Smad2, Smad3, and Smad4 genes. Moreover, the mTOR-dependent pathway was found to be altered by the anomalous expression of the Sos, Ras, PI3K, and PKB-Akt genes. Finally, the p53 was affected through a misleading expression of its gene. Taken together, these data suggested that our miRNAs may affect the tumoral development in colorectal cancer and involve the nervous system.

### 2.6. SCARA5 and SST Genes Were Related to Survival Prognosis in CRC

To investigate the relevance of the miRNA-targeting genes in cancer survival, we matched survival-related DEGs identified from 362 RNA-seq of CRC patients with our miRNA targets shown in Figure 2A. These data indicated that SCARA5 and SST genes were associated with patient survival. Specifically, SCARA5 mRNA upregulation was significantly associated with reduced survival (from 77% to 53%) of CRC patients at 22 months from diagnosis (*p*-value = 0.034) (Figure 5A). Also, SST mRNA upregulation significantly correlated with decreased survival rate (from 75% to 56%) at 21 months from the diagnosis of CRC (*p*-value = 0.044) (Figure 5B). We identified that hsa-let-7d-5p targeted both Scavenger receptor class A member 5 (SCARA5) and Somatostatin receptor 2 (SSTR2). Furthermore, hsa-miR-145-5p regulated SSTR1, 3 and 4 isoforms, and hsa-miR-331-3p targeted SSTR1 and 3. These results suggested that hsa-let-7d-5p, hsa-miR-331-3p, and hsa-miR-145-5p were involved in the survival of CRC through the regulation of SCARA5 and SSTR genes.

### 2.7. Tissue and Serum mRNA and miRNA Database Validation Led to SOCS1 mRNA, hsa-miR-331-3p, and hsa-let-7d-5p Candidates

We validated the reduced expression of hsa-miR-122-5p, hsa-let-7d-5p, and hsa-miR-223-3p and the upregulation of has-miR-150-5p in CRC compared to normal blood samples by investigating the dbDEMC (database of Differentially Expressed MiRNAs in human Cancers) (see Appendix A). However, to assess the application of our miRNAs of interest as possible biomarkers in CRC progression from UC, we identified the DEmiRs in human colon biopsies from UC and UC-associated CRC patients from GSE68306 RNA-seq data. We compared our serum miRNAs to the DEmiR profile, and we took into consideration only those miRNAs that were upregulated in both serum and tissue. The results showed that hsa-miR-331-3p was upregulated in both serum and tissue in UC versus CRC. This miRNA showed an FC = 2.3 in tissue (adj. *p*-value = 0.048), whereas in serum the FC = 3.09 (adj. *p*-value < 0.001). In parallel, for validating the previously reported mRNAs identified in silico as miRNA targets, we analyzed the UC-CRC-related biopsy patient cohort GSE10714 dataset. The DEGs identified in the GSE10714 dataset were matched with our in silico-characterized miRNA-mRNA targets and a total of 20 mRNAs were downregulated (see Appendix A). Then, we determined whether our validated miRNA targeted them. The results showed that hsa-miR-331-3p regulated SOCS1 mRNA (Figure 2B). Interestingly, SOCS1 is also a target of hsa-let-7d-5p. Collectively, these data indicated that hsa-miR-331-3p, SOCS1 mRNA, and indirectly hsa-let-7d-5p, could be considered promising biomarkers for UC-associated CRC diagnosis.

## 3. Discussion

UC has been linked to increased risk of developing CRC [13]. The recommendation for UC patients is surveillance colonoscopy with biopsy every 1 to 2 years [14]. Knowledge of prognostic factors of UC progression in CRC is important for classifying subgroups of UC patients who need frequent surveillance or intensive treatment. The aim of our study was to identify, by miRNA profiling and subsequent bioinformatic validation, potential biomarkers of UC disease progression to colorectal cancer. Some microarray-based studies have identified gene expression profiles in UC and CRC [15]. Long-term inflammatory bowel disease increases the risk of CRC; thus, we enrolled patients with a UC disease for at least 10 years [16]. We identified, by microarray analysis, 10 miRNAs (5 upregulated and 5 downregulated) as serum candidates of progression prediction from UC to CRC. We designed a ceRNA regulatory network to explore potential mechanisms regulated by miRNA candidates. Our attention was focused only on the highly interconnected upregulated miRNAs. Recent findings have confirmed that these miRNAs are related to inflammation and cancer, and some of them are directly involved in IBD and/or CRC disease. In detail, hsa-let-7d-5p has been found to be upregulated in different gastric cancer cell lines [17] and increased in serum from patients with inflammatory disease, being involved in neuronal development and immune-inflammatory response [18,19]. Another study highlighted a possible protective role of hsa-let-7d-5p by inhibition of the Wnt/catenin pathway in CRC [20]. Although hsa-let-7d-5p seems to be downregulated in CRC progression, this miRNA is upregulated in serum from patients with inflammatory diseases. Although hsa-miR-16-5p is widely reported as an important cancer suppressor, high levels of this miRNA have been correlated with poor survival in CRC [21,22]. Moreover, hsa-miR-16-5p has been found to be upregulated in the serum, tissue, and fecal samples of UC patients [23,24]. We reported for the first time hsa-miR-145-5p as a regulator of inflammatory processes. Moreover, its role in CRC development seems to be linked to tumor suppression [25]. By contrast, hsa-miR-223-3p is already recognized as upregulated in both serum from UC and biopsies from CRC patients [26,27]. Finally, hsa-miR-331-3p appeared to protect against neuronal-related inflammation, although other studies reported that an inhibition of this miRNA could ameliorate Alzheimer’s disease [28,29,30]. Interestingly, hsa-let-7d-5p, hsa-miR-223-3p, hsa-miR-145-5p, and hsa-miR-16-5p all regulated ACVR2A mRNA. ACVR2A has been shown to be a tumor suppressor in CRC [31]. Other important genes that form part of this network are ELK4 (miR-145-5p and let-7d-5p) and POU2F1 (miR-16-5p and let-7d-5p). Specifically, ELK4 inhibition in CD8^+^ cells seemed to contribute to CRC progression by favoring the immune microenvironment [32]. In contrast, the deletion of POU2F1 transcription factor revealed an improvement in cell homeostasis but blocked the recovery of the tissue in a colitis model. Moreover, POU2F1 partial inhibition, by the loss of one of its alleles, was correlated with colon tumorigenesis [33]. Finally, hsa-let-7d-5p, hsa-miR-16-5p, hsa-miR-145-5p, and hsa-miR331-3p were correlated with several important genes’ regulation such as SOCS1, IGF2BP1 (hsa-miR331-3p and hsa-let-7d-5p), FAM126A (hsa-miR-145-5p and hsa-miR331-3p), and CCDC85C (hsa-miR331-3p and hsa-miR-16-5p). The hypermethylation of the suppressor of SOCS1 was related to colitis-associated colorectal cancer progression [34]. IGF2BP1 contributed to aggressive phenotypes in CRC cell lines and was a poor prognostic marker in CRC patients [35]. FAM126A contributes to worse progression of pancreatic cancer by ENO1, a key activator of the PI3K/AKT signaling pathway [36]. Yu et al. found low levels of CCDC85C in a tumor murine model of CRC compared to a healthy one. Therefore, a lower expression of CCDC85C was a predictor of poorer survival in CRC patients [37]. Many studies confirmed differential expression of circRNAs between normal colon and CRC cell lines [38,39]. Analysis of a ceRNA sub-network showed that circRNA-SHPRH and circ-CEP128 regulated the expression of hsa-miR-331-3p and hsa-miR-145-5p, respectively. The circRNA-SHPRH was shown to be a promising biomarker in CRC progression [40], whereas the circ-CEP128 was correlated with bladder cancer [41]. Moreover, it was reported that both circ-ABCC1 and circ-CSNK1G1 targeted miR-145, inducing proliferation and invasion of CRC [42,43]. Therefore, among all the molecular targets of our miRNA candidates, POU2F1, SOCS1, and circ-SHPRH are better characterized for their involvement in colitis-associated CRC. All these targets are commonly regulated by hsa-miR-331-3p, suggesting the importance of the latter as the most promising biomarker in UC-CRC progression. In the PPI network, the most interrelated clusters included VEGF, SMAD, and TP53 signaling. The TP53 cluster included other genes affected in colon cancer, such as BCL2 and SOCS1. Another important cluster included SMAD, IGFR1, FOXO1, and PIK3CA. VEGFA, NRAS, CASP3, and AKT3 were all members of the same cluster. All these clusters highly correlate with CRC development. As we showed in the functional analysis, these pathways were listed in the top 10 relevant pathways. The top 10 genes in the PPI network were TP53, CASP3, VEGFA, IGF1, SMAD2, SMAD4, TGFBR1, THBS1, COL1A1, and PIK3CA. As we expected, our miRNA candidates seemed to impair TP53 activity, whose loss of function has been characterized in earlier states of colitis-associated CRC. In addition, CASP3 is a protein related to intestinal homeostasis, helping to maintain epithelial tight junctions. In IBD patients, the inhibition of CASP3 expression correlates with resistance to apoptosis which, in turn, can favor tumor progression [44]. Moreover, IGF1 is also linked to pro-tumor metabolic rearrangements, thus promoting CRC progression [45]. Although IGF1 was found to be upregulated in CRC, it was not reported in IBD. Indeed, higher levels of IGF1 are linked to the maintenance of the epithelial intestinal cell barrier in murine colitis [46]. In the case of the SMAD protein family, different pathogenic mutations were identified in CRC. These mutations reduced the activity of SMAD2-4 proteins, which was associated with the development of a mucinous phenotype, whose frequency is higher in colitis-associated CRC [47,48]. PIK3CA is mutationally activated in several tumors and IBD [49]. VEGFA and COL1A1 are both related to metastasis development and advanced CRC stage, but they appear to be downregulated by our miRNA candidates. This is likely due to the involvement of miRNA candidates in the early phases of cancer development [50,51]. The functional analysis showed the regulation of the SMAD axis, of several biological processes such as the development and regulation of the serine/threonine pathways. Therefore, tumor processes such as cell proliferation, cell and embryonic development, and regulation of apoptosis were all affected. On these bases, our data surprisingly showed that nearly all the top ten cell components and biological process terms correlated with neuronal development. Based on these data, it can be hypothesized that tumor development is a general process that also involves the neuronal cancer microenvironment and neo-neurogenesis. The top ten identified pathways also correlated with different tumor processes such as chronic myeloid leukemia, pancreatic, gastric, and colorectal cancer. Using a miRWalk machine learning algorithm, we analyzed the association between gene expression and prognosis of CRC patients. We found that hsa-let-7d-5p targeted both SCARA5 and SSTR. Also, hsa-miR-145-5p and hsa-miR-331-3p regulated different receptor SSTR isoforms. In fact, our miRNA candidates are predicted regulators of SSTR expression: hsa-let-7d-5p of SSTR2, hsa-miR-145-5p of SSTR1-3-4, and hsa-miR-331-3p of SSTR1-3. Furthermore, SST upregulation was significantly correlated with a decreased survival (from 75% to 56%) at 21 months from the diagnosis of CRC (*p*-value = 0.044). All these data strongly suggest a role of the colon neuroendocrine compartment in the determination of the progression from UC to CRC. SCARA5 gene expression was also identified as a survival biomarker in CRC patients. Some studies reported that SCARA5 was downregulated in several tumors, being involved in metastasis inhibition. In breast cancer, SCARA5 blocks ERK1/2, Akt, and STAT3 pathway activities [52]. However, its expression in CRC is not often studied, and additional research is required. SCARA5 upregulation was significantly associated with reduced survival (from 77% to 53%) of CRC patients at 22 months from diagnosis (*p*-value = 0.034).

## 4. Materials and Methods

### 4.1. Clinical Samples

Thirty patients were enrolled between January 2020 and March 2023 at the Hepatogastroenterology Division of the University of Campania “Luigi Vanvitelli” with pre-cancerous or cancerous intestinal tract diseases, specifically UC, CRC, and noninflammatory diseases. All participants have provided written informed consent. This study was approved by the ethical committee of the University of Campania “Luigi Vanvitelli”.

### 4.2. Sample Processing

A serum blood sample was collected from all enrolled patients. For serum collection, blood was centrifuged at 3000 rpm for 5 min at room temperature. After that, supernatant was collected and stored at −80 °C for a subsequent total RNA extraction.

### 4.3. RNA Purification

Total RNA, including small RNAs, was extracted from each serum blood sample using TaqMan^®^ ABC miRNA extraction kit (Applied Biosystems, Thermo Fisher, Carlsbad, CA, USA) according to the manufacturer’s protocol. Ath-miR-159a miRNA was added as an external control. Extracted miRNA was validated by ath-miR-159a retro-transcription (RT) and consequently cDNA amplification using TaqMan^®^ MicroRNA Reverse Transcription Kit Applied Biosystems by Thermo Fisher (CA, USA) according to the manufacturer’s protocol. Real-time quantitative PCR was performed on ViiA7™ Real-time PCR by Applied Biosystems using Fast 96-well format function in ViiA™ 7 Software. cDNA and amplified product were synthesized with TaqMan^®^ microRNA Assay INV SM10 ath-miR159a by Thermo Fisher (CA, USA).

### 4.4. miRNA Profiling and Real-Time PCR Analysis

In the first step, total RNA extract is reverse-transcribed using MicroRNA Reverse Transcription kit by TaqMan^®^ Applied Biosystems by Thermo Fisher (CA, USA) according to the manufacturer’s protocol. cDNA was synthesized with Megaplex RT primers, Human Pool A v2.1 Applied Biosystems by Thermo Fisher Scientific (CA, USA). cDNA was stored at −20 °C. During the second step, each RT group containing cDNA template was analyzed using TaqMan^®^ Array miRNA Human A Card Applied Biosystems, which offers the convenience of pre-spotted TaqMan Advanced miRNA assays (Thermo Fisher, CA, USA) according to the manufacturer’s protocol. Real-time quantitative PCR was performed on ViiA7™ Real-time PCR Applied Biosystems Applied Biosystems by Thermo Fisher (CA, USA). Relative expression (Ct value) of the transcripts was measured by using the 384-well TaqMan^®^ Array block function in ViiA™ 7 Software.

### 4.5. Statistical Analysis

Mean Ct values of miRNAs in each group were normalized to ∆Ct by using hsa-miR-483-5p as an internal control miRNA. For estimating the miRNA relative expression, the ∆∆Ct method and Fold Change (FC) calculated by 2^−∆∆Ct^ method were used. Only miRNAs that showed an FC ≥ |2| were further considered. In the PCR array screening phase, miRNA differential expression analysis was conducted with the limma Bioconductor package (https://bioconductor.org/packages/release/bioc/html/limma.html (accessed on 21 March 2024)) using linear regression models with empirical Bayes moderated t statistics. FDR-adjusted *p*-values were calculated using the Benjamini–Hochberg method. Statistical analysis, including the hierarchical clustered heatmap, was performed using GraphPad Prism Version 8 and gplots package from R 4.2.1.

### 4.6. miRNA Target Prediction

Target genes, mRNAs, of all significant previously identified miRNAs were predicted by TargetScan and miRDB databases, as well as by machine learning miRWalk methodology. Only genes found to be commonly represented in all three databases were selected as accurate miRNA targets. To predict miRNA-targeted long noncoding RNAs (lncRNAs), DIANA-LnBase version 2 and InterRNA databases were used, and the results were compared on the BioMart online platform. Only common targeted lncRNAs were considered for further analysis. In order to identify the possible circRNA-miRNA interactions of our miRNA candidates, we used the circBase web interface and Cancer Specific CircRNA Database (CSCD) version 2.0. Only common targeted circRNAs on both platforms were chosen as eligible molecules.

### 4.7. Competitive Endogenous RNA (ceRNA) ncRNA-miRNA-mRNA Network Establishment

The CeRNA network allowed us to analyze the targeting relationships. It was constructed to distinguish upregulated from downregulated miRNA profiles and visualized by Cytoscape v 3.9.1. A ceRNA subnetwork was built, for simplifying data. In all cases, an organic layout was used from the yFiles plugin (https://www.yworks.com/products/yfiles (accessed on 21 March 2024)).

### 4.8. Protein–Protein Interaction (PPI) Network Construction, Protein Clusters, and Hub Gene Analysis

A PPI network was constructed using upregulated miRNA-mRNA targets using STRING plugin in Cytoscape. Protein clusters from the PPI network were characterized through the MCODE plugin (degree = 5, node score = 0.2, k-core = 2) in Cytoscape. Finally, the top 10 hub genes were identified by the Maximal Clique Centrality (MCC) local-based method using the CytoHubba plugin in Cytoscape. The MCC method was shown to perform better in essential protein capture than the other 10 proposed by CytoHubba [40]. Only clusters with degree = 5, node score = 0.2, k-core = 2 were selected. All studies were conducted in Cytoscape 3.9.1. Sphere layout from the Cy3D plugin altogether with manual clustering was performed in order to visualize the main interaction between protein–protein clusters and other relevant PPI network molecules.

### 4.9. Enrichment Analysis of PPI Network

GO annotation, including biological process, molecular function, and cell component, was analyzed for PPI network data. Biological pathways were analyzed by the Kyoto Gene and Genomic Encyclopedia (KEGG). The functional analysis was conducted using the clusterProfiler package for groupGO, enrichGO, and enrichKEGG R 4.2.1. Only the top 10 elements with a *p*-value < 0.05 were considered significant.

### 4.10. Gene Expression Dataset

The GSE10714 RNA-seq dataset, publicly available at the Gene Expression Omnibus (GEO) website (https://www.ncbi.nlm.nih.gov/geo/ (accessed on 21 March 2024)), was downloaded. This dataset contains RNA-seq data of UC, CD, and CRC of 33 different frozen tissue biopsies. All analyses were performed in R 4.2.1, using the affy, affyPLM, limma, annafy, and gplots R packages. According to the quality control scanning, only 3 UC and 7 CRC samples were eligible for the differential expression study. A linear modeling approach was performed for upregulated and downregulated differentially expressed gene (DEG) identification. Genes with an estimated FC ≥ |2| were identified as interesting DEGs for further analysis.

### 4.11. In Silico Tissue–Serum DEG Profile Comparison

The previously identified miRNA target genes and the DEGs obtained from GSE10714 analysis were merged by creating a Venn diagram with a bioinformatic tool available at InteractiVenn (http://www.interactivenn.net/ (accessed on 21 March 2024)).

### 4.12. Identification of DEGs Related to CRC Survival Prognosis

For the validation and identification of possible genes involved in tumoral survival rates, TCGA data were downloaded and analyzed by R 4.2.1. Specifically, the TCGAbiolinks, limma, edgeR, glmnet, factoextra, FactoMineR, caret, SummarizedExperiment, gplots, survival, survminer, RColorBrewer, gProfileR, genefilter R packages were used. Within the TCGA data, we analyzed the COAD and READ datasets. DEGs were identified using linear modeling. We merged these identified DEGs with our miRNA-mRNA targets by constructing a Venn diagram on the InteractiVenn web interface. Only common genes were considered for survival prognostic tests by using the Kaplan–Meier estimator. Genes with a *p*-value < 0.05 were considered as CRC prognostic candidates.

### 4.13. miRNA Expression Dataset Analysis

The GSE68306 RNA-seq dataset, publicly available on the GEO website, was analyzed on the GEO2R web interface (https://www.ncbi.nlm.nih.gov/geo/geo2r/ (accessed on 21 March 2024)) using the linear regression modeling approach and estimating FDR-adjusted *p*-values by the Benjamini–Hochberg method. Only adjusted *p*-values < 0.05 were considered. This dataset contains the Nanostring nCounter System v1 miRNA array data analysis of RNA samples extracted from formalin-fixed paraffin-embedded tissue of controls (n = 12), UC without neoplasia development (n = 9), UC-associated neoplastic tissue (n = 10), and non-tumoral neighboring tissue (n = 9). In our analysis, we compared the differential miRNA expression of UC without neoplasia and UC-associated neoplastic tissue groups. Differentially expressed miRNAs (DEMiRs) were categorized into upregulated and downregulated miRNAs in the tumor status compared to the non-tumor UC.

### 4.14. In Silico Tissue–Serum miRNA Validation

The previously reported upregulated and downregulated identified miRNAs in both serum and tissue were merged by creating a Venn diagram. Serum miRNAs were obtained by our microarray analysis, whereas tissue miRNAs were characterized from GSE68306 RNA-seq bioinformatic analysis. Only those miRNAs that showed the same expression pattern in both serum and tissue were considered as validated miRNAs.

## 5. Conclusions

In our study, we identified a miRNA signature that is significantly associated with UC progression to CRC. The signature of hsa-miR-7d-5p, hsa-miR-16-5p, hsa-mir-145-5p, and hsa-miR-223-3p was associated with a common predicted mRNA target: ACVR2A.

Moreover, hsa-let-7d-5p and hsa-miR-331-3p regulated SOCS1 mRNA, an essential gene in CRC progression. Hsa-miR-331-3p and hsa-miR-145-5p were sponged by circ-SHRPH and circ-CEP128, respectively.

In conclusion, we propose two miRNAs, hsa-miR-331-3p and possibly hsa-let-7d-5p, as novel serum biomarkers for predicting UC progression to CRC. Both miRNAs targeted SOCS1 mRNA and were involved in CRC survival through the regulation of SCARA5 and SSTR genes. Our research is based on the combination of experimental and bioinformatic data. However, the analysis was performed on a small group of patients and should be confirmed on a larger cohort to be validated. Moreover, we will also study the effects of the modulated miRNAs on cell models of CRC both in vitro and in vivo.

## Figures and Tables

**Figure 1 ijms-25-05699-f001:**
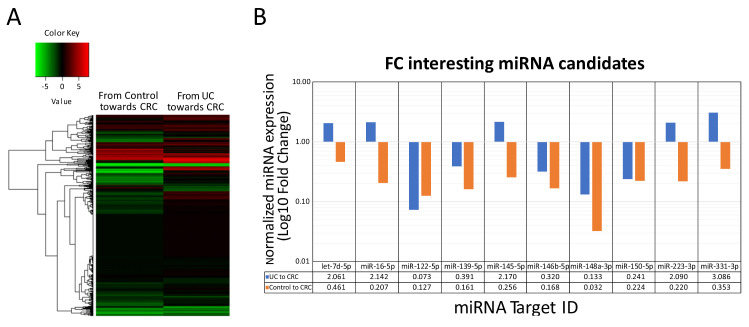
Microarray low-intensity profile analysis between UC and nonrelated IBD progression to CRC. (**A**) Global and differential expression levels of miRNA across the screening set as shown by hierarchical clustering heatmap. The analysis summarizes the differential patterns of miRNA expression across the groups, ulcerative colitis (UC) and nonrelated IBD inflammation control miRNA expression profiles compared to colorectal cancer (CRC) miRNA expression pattern. Reddish values indicate upregulation, whereas greenish values show downregulation; (**B**) logarithm in base 10 FC of our miRNA candidates comparing UC and control to CRC progression.

**Figure 2 ijms-25-05699-f002:**
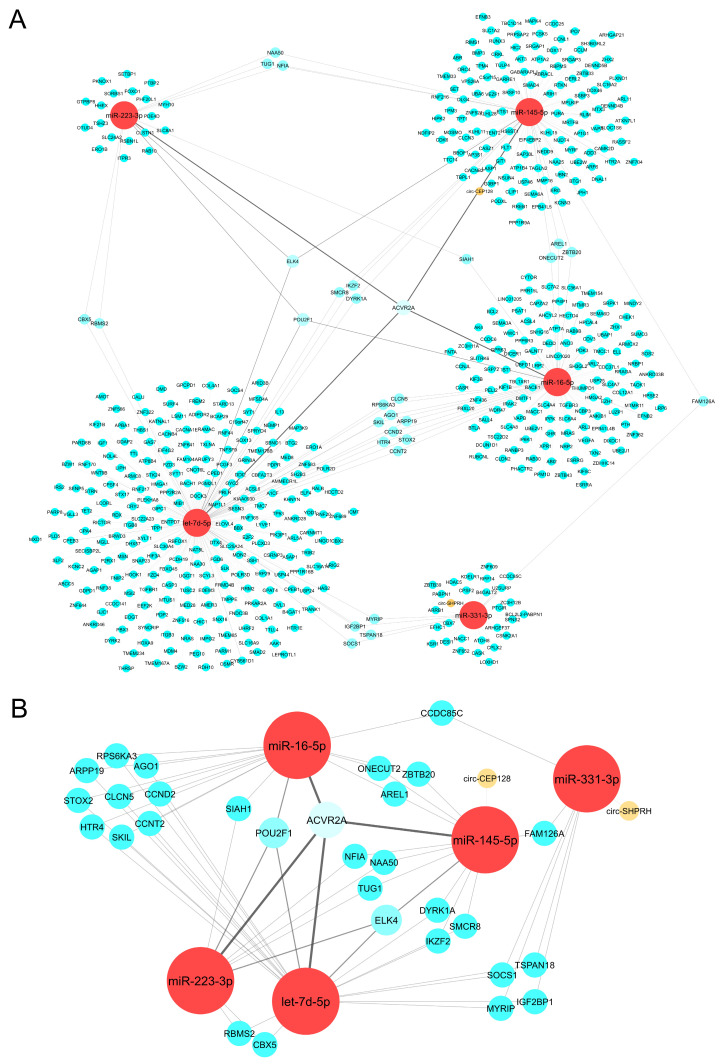
Upregulated miRNA candidates’ interaction with their inhibitors (circRNAs) and targets (mRNAs). (**A**) CeRNA circRNA-miRNA-mRNA links. A detailed description of the ceRNA members is provided in the Appendix A; (**B**) ceRNA subnetwork for upregulated miRNAs. Only the most significant interactions (3 or more links to other molecules) are represented in this subnetwork, in addition to the circRNAs. miRNAs are shown in red; mRNA targeted by different miRNAs (blue) showing 1 (small node size), 2 (medium node size), and more than 3 (bigger node sizes); circRNAs are shown in orange.

**Figure 3 ijms-25-05699-f003:**
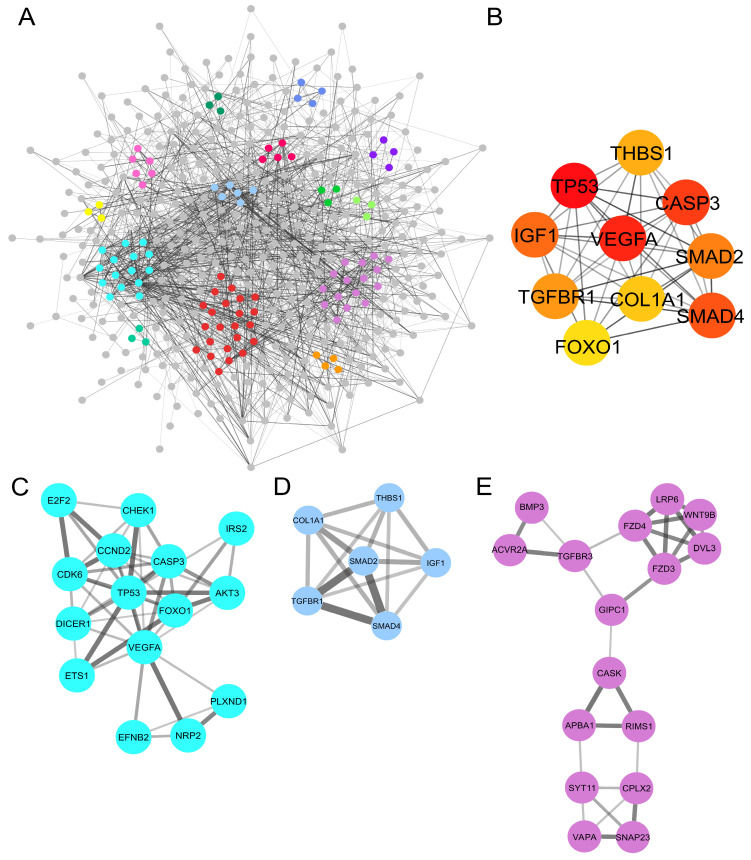
PPI network and its top ten hub genes constructed by miRNA targets. (**A**) PPI network built by using upregulated miRNA-mRNA targets; in total, 453 nodes and 1216 interactions between these nodes were established. In color, the different identified protein clusters are shown in the Appendix A; (**B**) top ten hub genes analyzed from the upregulated miRNA targets; (**C**,**D**) the most significant genes regarding the number of their interactions with other genes; (**E**) the main clusters in which our top hub genes were identified next to the ACVR2A protein of interest.

**Figure 4 ijms-25-05699-f004:**
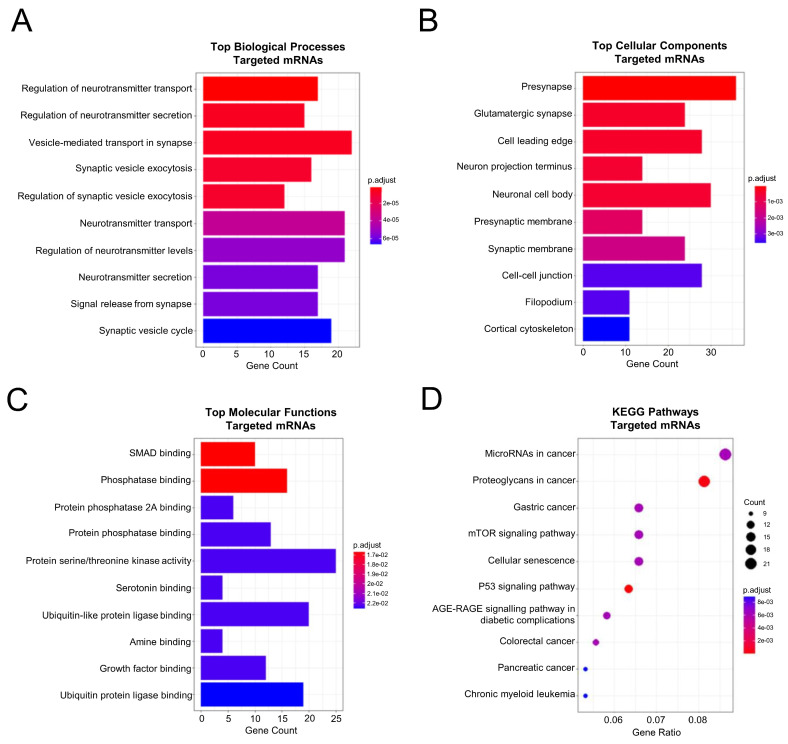
GO and KEGG pathway analysis of the upregulated miRNA targets. The GO top 10 terms of the biological processes (**A**), cellular components (**B**), and molecular functions (**C**). The length of each box correlates with the number of genes involved in each process, whereas the color is related to the significance of the value; (**D**) top 10 most significant KEGG pathways identified. The size of the circles represents the number of genes involved in each one of the pathways, whereas the color represents the significance of the value.

**Figure 5 ijms-25-05699-f005:**
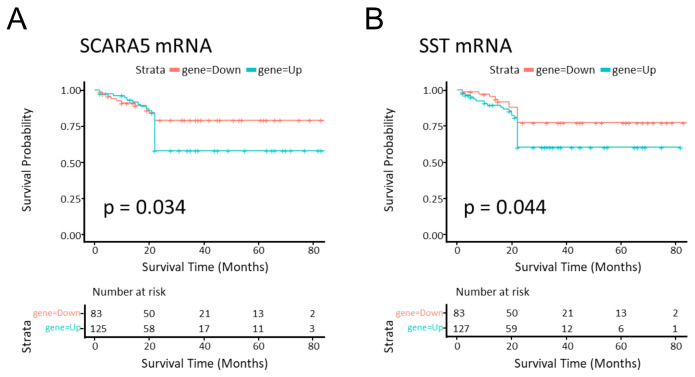
Survival analysis of SST and SCARA5 mRNA expression. Downregulated mRNA expression is represented in red, while upregulated expression is represented in blue. The boxes describe the number of patients analyzed in each period. Each cross shows when a patient was excluded from the analysis. (**A**) SCARA5 mRNA upregulation was also linked to a worse tumoral prognosis (*p*-value = 0.034); (**B**) SST mRNA upregulation was significantly related to a poorer prognosis (*p*-value = 0.044).

**Table 1 ijms-25-05699-t001:** Clinical information of all the patients involved in the microarray experiments.

Parameter	Control	UC	CRC	Overall
	(N = 10)	(N = 10)	(N = 10)	(N = 30)
Age (y)	5 (50%)	5 (50%)	5 (50%)	15 (50%)
<60 y	5 (50%)	5 (50%)	5 (50%)	15 (50%)
≥60 y	60.5	55.5	59.5	59.5
Overall *	(40–64.5)	(41.25–64.25)	(57–63.25)	(49.25–64)
Sex				
Female	5 (50%)	5 (50%)	5 (50%)	15 (50%)
Male	5 (50%)	5 (50%)	5 (50%)	15 (50%)
Disease Duration (UC) *	N/A	20	N/A	N/A
		(10.75–32.5)		
Disease Stage	N/A	N/A		N/A
I			3 (30%)	
II			3 (30%)	
III			2 (20%)	
IV			2 (20%)	

* Continuous data are expressed as median (interquartile range) and categorical/orderly variables as numerosity (percentage of total). UC: ulcerative colitis; CRC: colorectal cancer; N/A: not applicable.

## Data Availability

All data generated or analyzed during this study are included in this article and its Appendix A.

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
