# Peer review of "A Combination of Microarray-Based Profiling and Biocomputational Analysis Identified miR331-3p and hsa-let-7d-5p as Potential Biomarkers of Ulcerative Colitis Progression to Colorectal Cancer"

_ijms, 2024, doi:10.3390/ijms25115699_

Round 1

Reviewer 1 Report

Comments and Suggestions for Authors

Dear authors,

Congratulations for writing this manuscript.

It was an interesting read, but requires a few edits before publication can be considered:

1. Make sure you format your manuscript correctly:

- the affiliations are not correctly written

- the number of keywords are too many and make sure you only include Mesh keywords

- the subheadings are not properly formatted in all sections of the manuscript. 
- tables and figures should be cited in text before they first appear. Same for the abbreviations

2. References should follow the MDPI style, while your do not follow it. Please refer to the instructions for authors guidelines. 
3. I would include a focus on study limitations in the discussion section.

4. Please include a paragraph in the discussion section where you discuss the clinical utility of your findings.

Good luck! 

Comments on the Quality of English Language

Minor English grammar and spelling edits are required.

Author Response

We have mostly appreciated the comments of the referees and we have tried to follow their     suggestions to ameliorate the manuscript. We believe that their suggestions had a pivotal role in giving a clearer picture of the focus described in the present review manuscript. Therefore, we have modified the manuscript according to the referee suggestions as described below.

Response to Reviewer 1

Point 1.  Make sure you format your manuscript correctly:

- the affiliations are not correctly written

- the number of keywords are too many and make sure you only include Mesh keywords

- the subheadings are not properly formatted in all sections of the manuscript. 
- tables and figures should be cited in text before they first appear. Same for the abbreviations

OUR REPLY: We have corrected the affiliation, the number of keywords, and formatted the subheadings.

Point 2. References should follow the MDPI style, while your do not follow it. Please refer to the instructions for authors guidelines. 

OUR REPLY: We have corrected the mistakes on the reference.

Point 3. I would include a focus on study limitations in the discussion section.

OUR REPLY: We thank the reviewer for pointing out this important issue. In the conclusion section we have add the sentence: “Our research is based on the combination of both experimental and bioinformatic data. However, the analysis was performed on a small series of patients and should be confirmed on a larger cohort to be validated. Additionally, the biological effects of the modulated miRNAs should be addressed on cell line and, possibly, in vivo models of UC and CRC”.

Point 4. Please include a paragraph in the discussion section where you discuss the clinical utility of your findings.

OUR REPLY:  We thank the reviewer for pointing out this important issue. We have added in the discussion section this sentence: “The identification of potential predictive biomarkers, evaluable in patient sera, is very important to classify subgroups of UC patients that require continuous follow-up or more intensive chemopreventive treatment to avoid CRC development.”

Reviewer 2 Report

Comments and Suggestions for Authors

This work by Chacon-Millan and colleagues studies the expression of a panel of microRNAs in serum from patients suffering from ulcerative colitis (UC), colorectal cancer (CRC) and control individuals showing nonrelated inflammatory bowel disease. Based on the obtained profile, a series of in silico analyses are performed to identify networks of genes and pathways targeted by the selected microRNAs, leading to the conclusion that miR331-3p and has-let-7d-5p could be considered as potential biomarkers of UC progression to CRC.

It represents and interesting and deep in silico study of potential clinical application. However, it needs something more to reach real relevance. I hope my comments help in this way.

MAIN COMMENTS

1.        Few patients were enrolled for the microarray-based profiling with no further validation. At least, a validation of the top 10 differentially expressed miRNAs should be performed in a larger validation cohort. This is imperative to validate the proposed markers.

2.        Once the validation is performed, sensitivity and specificity studies should be developed to understand real performance of these markers.

3.        Statistical analyses performed to address the fold change (FC) in tissue and serum in the analysis performed from GSE68306 RNA-seq data should be indicated. Is this FC statistically significant?

4.        Line 16: Ester Pagano is not listed as an author.

MINOR

5.        Line 29. “were to be linked”. Is that correct?

6.        Lines 105 – 106. “in CRC patients progressed from UC disease”, is that correct? Can the authors assure that the CRC patients enrolled developed their cancer from UC. If not, this should be corrected. If so, this should be stated, at least, in the description of the clinical samples (section 4.1.)

7.        Figures show a really poor quality. This should be improved to facilitate the reading.

8.        Line 153: “clustered?”. Is that correct?

9.        Lines 216 -220: Text describing figure 5A indicates figure 5B and the other way around. Please, correct.

10.   Lines 258 – 259. Does really reference 17 show “Some microarray-based studies have identified gene expression profiles in UC and CRC”? This is relevance because this sentence helps to understand the novelty of this work.

Author Response

We have mostly appreciated the comments of the referees and we have tried to follow their     suggestions to ameliorate the manuscript. We believe that their suggestions had a pivotal role in giving a clearer picture of the focus described in the present review manuscript. Therefore, we have modified the manuscript according to the referee suggestions as described below.

Response to Reviewer 2

MAIN COMMENTS

Point 1.        Few patients were enrolled for the microarray-based profiling with no further validation. At least, a validation of the top 10 differentially expressed miRNAs should be performed in a larger validation cohort. This is imperative to validate the proposed markers.

Point 2.        Once the validation is performed, sensitivity and specificity studies should be developed to understand real performance of these markers.

OUR REPLY: We thank the reviewer for pointing out this important issue. We have checked in different databases the expression of our identified miRNAs, unfortunately only few information is available on circulating miRNAs expression. However, by investigating dbDEMC (database of Differentially Expressed MiRNAs in human Cancers) we validated the reduced expression of the hsa-miR-122-5p, hsa-let-7d-5p and hsa-miR-223-3p in CRC compared to normal samples (Fig below, for reviewer's evaluation only). Regrettably, no data was found for the other target miRNAs. We do agree that absence of information related to the other miRNAs is a limitation to the study therefore we have now added the following sentence to the conclusion section: Our study, based on the combination of both experimental and bioinformatic data, has been performed on a limited number of patients, and validation on a larger patient cohort should be performed to improve the prognostic value of the identified miRNAs.

Point 3.        Statistical analyses performed to address the fold change (FC) in tissue and serum in the analysis performed from GSE68306 RNA-seq data should be indicated. Is this FC statistically significant?

OUR REPLY: We thank the reviewer for highlighting this essential point. The differential expression of both GSE68306 RNA-seq data, containing tissue biopsies, and our microarray data, containing serum samples, were analyzed using linear regression models with empirical Bayes moderated t statistics performed by limma package in R 4.2.1. FDR-adjusted p values were calculated using the Benjamini-Hochberg method. Hsa-miR-331-3p was upregulated in tissue (FC = 2.3, adj p value = 0.048) and serum (FC = 3.09, adj p value = 1.103e-10) from CRC vs UC patients.

Point 4.        Line 16: Ester Pagano is not listed as an author.

OUR REPLY: we corrected the mistake.

MINOR

  1. Line 29. “were to be linked”. Is that correct?

OUR REPLY: we corrected the mistake.

  1. Lines 105 – 106. “in CRC patients progressed from UC disease”, is that correct? Can the authors assure that the CRC patients enrolled developed their cancer from UC. If not, this should be corrected. If so, this should be stated, at least, in the description of the clinical samples (section 4.1.)

OUR REPLY: We corrected the sentence with: “CRC patients vs UC patients”

Point 7.        Figures show a really poor quality. This should be improved to facilitate the reading.

OUR REPLY: We added in the manuscript after the Reference section all the figures.

Point 8.        Line 153: “clustered?”. Is that correct?

OUR REPLY: we corrected the mistake.

Point 9.   Lines 216 -220: Text describing figure 5A indicates figure 5B and the other way around. Please, correct.

OUR REPLY: We corrected the mistake.

Point 10.   Lines 258 – 259. Does really reference 17 show “Some microarray-based studies have identified gene expression profiles in UC and CRC”? This is relevance because this sentence helps to understand the novelty of this work.

OUR REPLY: We corrected the reference 17 with: “Shahnazari, M., Afshar, S., Emami, M.H. et al. Novel biomarkers for neoplastic progression from ulcerative colitis to colorectal cancer: a systems biology approach. Sci Rep 13, 3413 (2023)”. 

Thank you again for your kind consideration of our work.

Best Regards,

Paola Stiuso

Round 2

Reviewer 2 Report

Comments and Suggestions for Authors

Thank you for considering my comments. Most of my concerns have been addressed. There are still some minor points.

Regarding response to points 1 and 2.

Thank you for the effort. I appreciate it. Having this in silico confirmation, at least for some of the miRNAs, it should be included in the manuscript, at least, as supplementary data.

Regarding response to point 3.

Great. These statistics should be included in the manuscript.

Regarding point 7.

Not sure if this has been solved. At least, not in the pdf version I have.

Author Response

We have mostly appreciated the comments of the referees and we have tried to follow their     suggestions to ameliorate the manuscript. We have modified the manuscript according to the referee suggestions as described below.

Regarding response to points 1 and 2.

Thank you for the effort. I appreciate it. Having this in silico confirmation, at least for some of the miRNAs, it should be included in the manuscript, at least, as supplementary data.

OUR REPLY: we have add in SM figure 3 the Identified circulating miRNAs in dbDEMC (database of Differentially Expressed MiRNAs in human Cancers).

Regarding response to point 3.

Great. These statistics should be included in the manuscript.

OUR REPLY: We have add in the section “Statistical Analysis“ the sentence:In the PCR array screening phase, miRNAs differential expression analysis was conducted with limma Bioconductor package using linear regression models with empirical Bayes moderated t statistics. FDR-adjusted p values were calculated using the Benjamini-Hochberg method.”

Regarding point 7.

Not sure if this has been solved. At least, not in the pdf version I have.

OUR REPLY: Sorry for mistake in the new version of the manuscript we have add the figure in tiff.

Best Regards,

Paola Stiuso

Prof.ssa Paola Stiuso, PhD

Department of Precision Medicine

University of Campania “L. Vanvitelli" Naples Italy

Phone: +390817635 (office)

Fax: +390815665863

email: paola.stiuso@unicampania.it
